# Effects of Feeding and Drinking Behavior on Performance and Carcass Traits in Beef Cattle

**DOI:** 10.3390/ani12223196

**Published:** 2022-11-18

**Authors:** Bianca V. Pires, Henrique G. Reolon, Natalya G. Abduch, Luana L. Souza, Leandro S. Sakamoto, Maria Eugênia Z. Mercadante, Rafael M. O. Silva, Breno O. Fragomeni, Fernando Baldi, Claudia C. P. Paz, Nedenia B. Stafuzza

**Affiliations:** 1Beef Cattle Research Center, Institute of Animal Science (IZ), Sertãozinho 14160-900, SP, Brazil; 2Department of Genetics, Ribeirão Preto Medical School, University of São Paulo (USP), Ribeirão Preto 140349-900, SP, Brazil; 3Department of Animal Science, School of Agricultural and Veterinary Sciences, Sao Paulo State University (UNESP), Jaboticabal 14884-900, SP, Brazil; 4Zoetis, Sao Paulo 04710-230, SP, Brazil; 5Department of Animal Science, University of Connecticut, Storrs, CT 06269, USA

**Keywords:** feeding rate, residual feed intake, residual water intake

## Abstract

**Simple Summary:**

Sustainability in livestock production includes the use of strategies to reduce natural resource requirements. In this study, we investigated the relationship among feed efficiency, water efficiency, ingestive behavior, performance, and carcass traits in beef cattle. The results revealed interesting aspects of both feed efficiency and water efficiency on ingestive behavior and growth traits. The combined use of residual water intake and residual feed intake is an important option available for improving the environmental sustainability of beef cattle production that could be used in animal breeding programs.

**Abstract:**

Feed and water efficiency are important traits to improve beef cattle production’s economic and environmental sustainability. This study evaluated residual feed intake (RFI) and residual water intake (RWI) and their relationship with performance, ingestive behavior, and carcass traits in Caracu beef cattle. The data were analyzed using a generalized linear model with least squares means. The ingestive behavior, performance, and carcass traits were influenced by sex (*p* < 0.05). Males showed higher dry matter intake (DMI), average daily gain (ADG), mid-test metabolic weight (BW^0.75^), rib eye area, and rump fat thickness than females, besides spending more time drinking and eating. Low RFI animals exhibited higher DMI than high RFI animals. Low RWI animals ingested 3.89 L/d of water further than high RWI animals. The interaction between sex and RWI influenced the DMI, BW^0.75^, and backfat thickness. The ingestive behavior of low and high RFI animals was similar, although high RWI animals visited a smaller number of drinkers than low RWI animals. Water intake positively affects productive efficiency, and the combined use of RWI and RFI may help improve the selection of more efficient animals contributing to reducing the costs of beef cattle production and improving environmental sustainability.

## 1. Introduction

The effects of climate change on water availability are a matter of growing concern that pressures livestock production for more efficient sustainable agricultural practices. Water intake (WI) is a poorly explored measure in livestock research, which is influenced by various factors such as climate, diet, and body weight, as well as the physiological state of the animal [1]. Studies with taurine cattle breeds showed that WI is positively correlated with average daily weight gain [2,3]. However, further studies on water intake and feed efficiency are needed to determine whether the former could be an indicator trait of feed efficiency.

Feed corresponds to a large part of the total costs of beef production and the adoption of selection criteria for feed efficiency by breeding programs has gradually increased around the world, making beef cattle production systems more profitable by reducing production costs and by increasing production [4]. In addition, improving feed efficiency has positive environmental impacts by reducing greenhouse gas emissions and solid waste [5,6]. Several feed efficiency measures have been proposed to better balance the relationship between feed intake and performance in beef cattle. Residual feed intake (RFI), proposed in the 1960s, is a widely used feed efficiency measure because it is considered to be independent of the growth and body size of animals. This trait is determined as the difference between the observed feed intake and the expected feed requirements for the maintenance of body weight and performance [7].

Studies have reported variations in feed efficiency phenotypes for animals of the same breed and under the same diet [1,8,9,10]. These findings may be explained by the fact that this trait is controlled by various physiological processes such as feed intake, digestion, metabolism, and thermoregulation [8]. The individual variation observed in feed efficiency can also be influenced by characteristics such as temperament, stress, appetite, ingestive behavior (water and food), oxygen consumption, energy waste, and thermotolerance [11,12,13]. According to Montanholi et al. [14], ingestive behavior plays an important role and could explain 18% of the total phenotypic variation for RFI.

Ingestive behavior traits allow for determining feeding patterns, which have been described to be strongly correlated with feed efficiency traits [15,16]. Positive and moderate phenotypic correlations between feeding behavior and RFI have been reported [5,9,17]. Studies have shown that high-efficiency animals (low RFI) spend less time on feed-related activities than low-efficiency animals (high RFI) [15,16,18].

A positive correlation between RFI and subcutaneous fat was observed in Nelore bulls [19]. Therefore, according to Santana et al. [19], more efficient animals for RFI may have lower subcutaneous fat deposition in the carcass. A study with Senepol bulls showed that low RFI animals had a lower rump fat thickness (RF), although the size reduction was not accompanied by a reduction in backfat thickness (BF) [16]. On the other hand, several authors have observed phenotypic correlation ranged 0.03 to 0.16 between RFI and carcass traits [11,20,21].

The variability in cattle feed efficiency traits has been described in the literature, but their association with water efficiency and ingestive behaviors was little explored. Thus, the aim of the study was to evaluate the effects of feed and water efficiency and behavior traits on both performance and carcass traits of Caracu (*Bos taurus taurus*), which has the largest effective herd among the Brazilian Creole breeds, in order to elucidate the relationship between efficiency and behavior traits and how it can affect carcass and performance traits.

## 2. Materials and Methods

### 2.1. Animals and Feed Efficiency Test

A total of 104 Caracu cattle (61 intact males and 43 females) from a single herd and the same birth season were evaluated in the feed efficiency test. The animals were allowed to adapt to the diet for 30 days before the beginning of the experiment. Water and feed were supplied with Intergado^®^ electronic feed bunks and drinkers (AF-1000 Master Gate, Betim, Brazil) which determines individual feeding behavior and feed intake in cattle with high specificity and sensitivity [22,23,24]. All animals were fitted with an ear tag containing a unique passive transponder, allowing free access to the feed bunk or water troughs.

The feed efficiency test for the males lasted from 31 May to 15 August 2019 at an initial age of 7.70 ± 0.75 months and an initial body weight (BWi) of 199.77 ± 4.77 kg. Females were assessed from 16 August to 30 October 2019 at an initial age of 10.36 ± 0.72 months and BWi of 246.23 ± 5.32 kg.

### 2.2. Traits

The diet (Appendix A) was formulated for 1200 kg/day to meet the maintenance and growth requirements of the animals using the RLM 3.3 software (ESALQ, Piracicaba, Brazil). The diet was offered twice a day (8:00 a.m. and 3:00 p.m.), and the volume was adjusted daily to maintain 10% of leftovers in all troughs. The animals were weighed before the beginning and after the end of the feed efficiency test, after fasting from feed and water for 16 h.

Diet samples were collected weekly. The samples were pre-dried for 72 h at 65 °C in a forced ventilation oven and ground in a knife mill (Thomas Scientific, Swedesboro, NJ, USA) using a 1-mm sieve for determination of the first dry matter (DM) content. Next, 2 g of each ground sample was dried at 105 °C in a forced ventilation oven and then weighed for determination of the second DM content (AOAC, Official Method 934.01, 1990). Samples of each diet ingredient were collected weekly to analyze DM content.

Dry matter intake (DMI) was calculated as an average of daily feed intake (71 days for males and 68 days for females), premultiplied by the weekly DM content of the diet offered. The average daily weight gain (ADG) was obtained using the following formula:(1)$$\gamma_{i\;}=\alpha +\beta *DOT_{i}+\varepsilon_{i}$$
where $\gamma_{i\;}$ is the weight of the animal in the $i^{th}$ observation; α is the intercept of the regression equation that corresponds to the initial weight; β is the linear regression coefficient that corresponds to ADG; $DOT_{i}$ is the days on test in the $i^{th}$ observation; and $\varepsilon_{i}$ is the random error associated with each observation. The mean mid-test metabolic body weight (BW^0.75^) was obtained for each period as follows: (2)$$BW^{0.75}={\left[\alpha +\left(\frac{1}{2}ADG*DOT\right)\right]}^{0.75}$$
where α is the intercept of the regression equation that corresponds to the initial weight, and DOT is the days on test. The RFI was obtained as proposed by Koch et al. [7]: (3)$$DMI=\beta_{0}+\left(\beta_{Bw^{0.75}}BW^{0.75}\right)+\left(\beta_{ADG}ADG\right)+\varepsilon$$
where DMI is the observed dry matter intake; $\beta_{0}$ is the intercept of the equation; $BW^{0.75}\;$ is the mean mid-test metabolic weight; $\beta_{Bw^{0.75}}$ is the regression coefficient of $BW^{0.75}$; ADG is the average daily gain; $\beta_{ADG}$ is the regression coefficient of ADG; and ε is the residual (RFI). The animals were classified into two categories based on the RFI values obtained: high-efficiency (RFI < 0) and low-efficiency (RFI > 0) animals.

Water intake (L/d) was measured daily and the residual water intake (RWI) was calculated as the residual of the linear regression equation of average WI on DMI and BW^0.75^ (RWI_DMI_), as described in Ahlberg et al. [25]. The animals were classified as high-efficiency (RWI < 0) and low-efficiency (RWI > 0) animals. 

The feeding rate (FR, kg/min) and drinking rate (DR, L/min) were evaluated and divided into four periods of the day: dawn (00:00 to 5:59 a.m.), morning (6:00 to 11:59 a.m.), afternoon (12:00 to 5:59 p.m.), and night (6:00 to 11:59 p.m.). The FR was calculated as the ratio between DMI and the time the animal spent at the feed bunk. The DR was calculated as the ratio between the volume of ingested water and the time the animal spent at the water trough. 

### 2.3. Ingestive Behavior and Carcass Traits

The following ingestive behavior traits were evaluated: drinking duration (DD, min/d), feeding duration (FD, min/d), number of feeders visited per day (NF, number/d), number of water troughs visited per day (NW, number/d), frequency of visits to the feed bunks with intake (FVF, visits/d), and frequency of visits to the water troughs with intake (FVW, visits/d). 

Studies suggest that the Intergado^®^ system is a useful tool for monitoring feeding and drinking behavior as well as water and feed intakes in young cattle [22,24]. The data on animal behavior were continuously recorded and transferred to the Intergado^®^ web software via a general packet radio service (GPRS). The system included a backup battery with up to five hours of energy when the main power fails. For each visit to the feed bunk or water through, the system recorded the animal number, feed bunk, or water through number, initial and final times, and weight to calculate the visit duration and the intake (feed or water).

At the end of the feed efficiency test, rib eye area (REA), BF, and RF traits were measured by ultrasound (Pie Medical 401347-Aquila, Esaote Europe BV, Maastricht, The Netherlands) using a 3.5-MHz linear probe (18 cm). The ultrasound images were analyzed using the Echo Image Viewer 1.0 program (Pie Medical Equipment BV, 1996). The REA was measured transversely in the *Longissimus thoracis* muscle between the 12th and 13th rib and is an indicator of carcass finishing and yield. The BF was also measured between the 12th and 13th rib and quantified subcutaneous fat thickness in the *L. thoracis* muscle, indicating the degree of carcass finishing. The RF was measured in the region between the intersection of the *Gluteus medius* and *Biceps femoris* muscles and indicates the degree of carcass finishing, associating growth precocity and finishing.

### 2.4. Statistical Analysis

The statistical analyses were performed using the SAS software (v. 9.4, SAS Institute Inc., Carrey, NC, USA). Pearson’s correlation coefficient between the variables studied was obtained with the CORR procedure. The daily mean values of the ingestive behavior traits were obtained using the summary procedure. 

The performance, behavior, and carcass traits were evaluated using the GLM procedure. The fixed effects of sex, RFI and RWI class, the interaction of RWI or RFI class with sex, and the covariate of age were used to evaluate DMI, WI, ADG, and BW^0.75^. The FR and DR were evaluated according to RFI class, RWI class, period of the day, sex, and interactions of sex with RWI, RFI, and period of the day.

The behavior traits at the feed bunk (NF, FVF, and FD) were analyzed considering the fixed effects of sex, RFI class, and their interaction. For the behavior traits at the water trough (NW, FVW, and DD), the model included the fixed effects of sex, RWI class, and their interaction. The model used for the carcass traits (REA, BF, and RF) included the fixed effects of sex, RWI class, and age as covariates.

## 3. Results

The mean DMI, ADG, and BW^0.75^ were 8.48 kg/d (6.06 to 10.35 kg/d), 1.01 kg/d (0.42 to 1.46 kg/d), and 66.38 kg (45.56 to 81.44 kg), respectively. Regarding carcass traits, the mean REA, BF, and RF were 57 cm^2^ (36.00 to 80.70 cm^2^), 2.46 mm (0.00 to 6.80 mm), and 4.56 mm (2.30 to 8.36 mm), respectively (Appendix A).

The RFI showed a significant positive and moderate correlation with DMI in both females (0.52) and males (0.58), a low correlation with BF (0.28) and FVW (0.26), and a moderate correlation with FVF (0.42) in males (Table 1). The carcass traits did not show a correlation with RFI or RWI in females, however, was observed a positive low correlation between RFI and BF (0.28), and a negative correlation between RWI and REA (−0.35) in males. The RWI and WI exhibited a positive correlation in both sexes (0.71 for females and 0.59 for males), whereas a negative correlation between RWI and BWi (−0.42) was observed in males (Table 1). 

The BWi is an important trait in animal evaluation, which demonstrated a high positive correlation with DMI (0.68 for males and 0.69 for females), WI (0.47 for males and 0.55 for females), BW^0.75^ (0.98 for both sexes), and REA (0.61 for males and 0.75 for females) in both males and females. The DMI presented a high and positive correlation with BW^0.75^ (0.74) and a moderate correlation with REA (0.53) in both sexes. Positive high correlations were also observed between BW^0.75^ and REA in both males (0.62) and females (0.75). The WI exhibited a positive and moderate correlation with BW^0.75^ in both sexes (0.60), whereas in males a moderate correlation with ADG (0.49) was observed, and a low correlation NW (0.28) and DD (0.27) (Table 1).

The NF and FVF ingestive behavior traits showed negative correlations with BWi in males (−0.26, −0.44) and with BW^0.75^ (NF: −0.26 for males and −0.42 for females; FVF: −0.46 for males and −0.37 for females). Regarding feed behavior in bunks, the animals with low BWi showed a disadvantage compared to the animals with high BWi, as expected. The FVW showed positive high correlations with NW (0.76) and DD (0.50) in both sexes, whereas positive moderate correlations were observed between NF and FVF (0.48 for males and 0.54 for females), and DD and FVW (0.51 for males and 0.33 for females). These results can be associated with competition behavior for food and water (Table 1).

The low RFI animals (*n* = 47) consumed 8.59 ± 0.09kg/d whereas high RFI animals (*n* = 57) consumed 7.87 ± 0.09kg/d on average, without significant differences observed in ADG (Table 2). The low RWI animals (*n* = 47) ingested 20.62 ± 0.34 L/d whereas high RWI animals (*n* = 57) ingested 19.91 ± 0.37 L/d, showing a significant difference in WI between RWI classes (*p* = 0.0001). 

Females and males presented different values of DMI, ADG, BW^0.75^, REA, and RF (*p* < 0.001). Males displayed higher DMI, ADG, BW^0.75^, REA, and RF. Low RFI animals had higher DMI than high RFI animals, with no significant difference observed for ADG. Females and males showed similar WI and BF (*p* > 0.05). The animals with low RWI ingested 3.89 L/d further than the animals with high RWI (*p* = 0.0001) (Table 2). The initial age of animals significantly influenced DMI, WI, ADG, BW^0.75^, and REA (*p* < 0.05) (Appendix A). The older animals at the beginning of the test had higher DMI and consequently higher ADG. The age of the animals at the measurement of the carcass traits did not significantly affect BF and RF. 

The interaction between sex and RWI influenced the DMI (*p* = 0.0206), BW^0.75^ (*p* = 0.0015), and BF (*p* = 0.0368) (Appendix A). Males and females with high RWI showed the highest DMI (9.26 ± 0.14 kg) and lowest DMI (7.13 ± 0.25 kg), respectively. The DMI of animals with high and low RWI were similar in males (9.26 ± 0.14 and 9.00 ± 0.14 kg). No significant difference was observed between females with low or high RWI (7.13 ± 0.25 and 7.53 ± 0.24 kg). 

The lower means of BW^0.75^ were observed in females (58.02 ± 2.03 kg for high RWI and 60.92 ±1.32 kg for low RWI) and showed different values (*p* < 0.05) in comparison with males (72.62 ± 1.18 kg for high RWI and 68.11 ± 1.19 kg for low RWI). The males with high RWI showed 4.53 kg more than males with low RWI (*p* = 0.0050) (Appendix A).

A higher BF was found in females with high RWI (3.34 ± 0.51 mm), following males with low RWI (2.44 ± 0.30 mm), females with low RWI (2.36 ± 0.33), and males with high RWI (2.23 ± 0.29 mm). No significant difference was observed between males and females with low RWI. The WI, ADG, REA, and RF were not influenced by the sex and RWI interaction (Appendix A). The traits studied were also not influenced by the sex and RFI interaction (Appendix A).

The FR was higher in females (*p* = 0.0001). The animals showed a higher FR in the morning (6:00 to 11:59 am), followed by the afternoon, night, and dawn periods (*p* = 0.001). High RFI animals had a higher FR than low RFI animals (*p* = 0.0152) (Table 3). 

The sex and RFI interaction influenced (*p* = 0.0197) the FR, where the highest FR was observed in females with high RFI, whereas the females with low RFI and males (both low and high RFI) did not show statistical differences in FR. No difference in FR was observed in sex and period interaction nor in sex and RWI interaction (Table 3). 

Males drank 0.15 L/min more water than females. Differences in DR were observed between sexes and periods of the day interaction (*p* = 0.0001), with the observation of a higher DR in males at night (1.19 ± 0.05 L) and a lower DR in females at dawn (0.73 ± 0.06 L). The DR did not differ between males and females in the morning or afternoon period. High RFI and high RWI animals had a lower DR than low RFI and low RWI animals, showing a difference of 0.13 L/min between classes. The interaction of sex and RFI or RWI did not influence the DR (Table 3).

The animals visited on average 2.87 different water troughs per day and 10.22 different feed bunks per day, spending 21.33 min per day drinking water and 160.15 min per day feeding. The FVW was 5.42 times per day, and the FVF was 58.5 times per day (Appendix A). Males and females differed in NF, FVF, and FD (Appendix A). Males visited fewer feed bunks-NF (9.94 ± 0.13 vs 10.59 ± 0.16 visits/d) and also had a lower FVF (53.53 ± 1.95 vs 64.77 ± 2.33 visits/d) compared to females. The FD was similar between sexes (162.37 ± 2.71 and 157.11 ± 3.23, *p* = 0.2152) (Appendix A). 

Animals with high RFI showed lower FVF than animals with low RFI (55.68 ± 2.26 vs 62.62 ± 2.04, *p* = 0.0247). The RFI did not influence the NF and FD behaviors (*p* > 0.05). The NF, FVF, and FD were not influenced by the interaction between RFI and sex (Appendix A). 

The NW was higher in females compared to males (2.75 ± 0.03 vs 3.03 ± 0.04 water stations day; *p* = 0.0001). Animals with low RWI visited a larger number of water troughs per day than animals with high RWI (2.83 ± 0.04 vs 2.96 ± 0.03 day; *p* = 0.0135). The FVW was higher in females compared to males (5.91 ± 0.15 vs 5.02 ± 0.11 visits/d; *p* = 0.0001). High RWI animals visited the drinkers on average 5.16 ± 0.15 times per day, whereas low RWI animals visited the drinkers 5.77 ± 0.12 times per day (*p* = 0.0022). No differences in DD were observed between males and females (20.46 ± 0.91 vs 22.99 ± 1.21 min/d; *p* = 0.0984) or between low and high RWI animals (22.50 ± 0.92 vs 20.95 ± 1.19 min/d; *p* = 0.3085) (Appendix A). The interaction between sex and RWI did not influence the behaviors at water troughs (*p* > 0.05) (Appendix A).

## 4. Discussion

Data collected on 104 animals were used to assess the effects of feed and water efficiency and behavior traits on performance and carcass traits. The results showed the relationship between efficiency and behavior measurements and how they affected affect carcass and performance traits.

Water participates in metabolic and digestive processes, as demonstrated by the moderate positive correlation of WI with BWi, DMI, and BW^0.75^ observed in the present study in animals of both sexes. Studies have shown that, for 3% of DMI, the animal consumes 10% of water, i.e., the WI is approximately three times the DMI [2,26,27,28]. The importance of WI is directly associated with the DMI and the health of the animal since water deprivation causes sudden changes in weight gain and consequently affects the health of animals [29,30]. 

The RWI was positively correlated with ADG in Caracu males, and similar results have been reported by Ahlberg et al. [31] in which an increase in feed intake also increased the WI in Angus cattle. Brew et al. [3] also found an association between WI and animal performance in several beef cattle breeds and crosses, in which animals that consumed more water had a higher body weight. The feed and water ingestion act together in the growth and development of animals, showing a positive correlation in our study (WI and DMI). 

The positive association of BWi, DMI, BW^0.75^, and FD with REA in both sexes confirms the importance of diet and nutrient utilization for meat cut yield and muscling degree in cattle [10,32]. Feed intake and WI are positively associated with the productive efficiency of the animal, and our study demonstrated a positive association between these traits, mainly in males. It is expected that animals with better efficiency in feed utilization exhibit a better degree of carcass finishing, as well as greater weight gain and growth [33], as indicated by the positive correlations of BW^0.75^ with RF and BF observed in the present study.

The DMI and ADG obtained for males are higher than those observed in Nellore animals at a mean age of 541 days (DMI = 6.74 kg/d; ADG = 0.74 kg/d) [34] and similar to those observed in Angus animals at a mean age of 322 days (DMI = 7.6 kg/d; ADG = 1.01 kg/d) [35]. These values indicate good performance of the Caracu animals used in this study when compared to taurine and zebuine breeds for meat production. 

The performance traits and carcass traits are influenced by the metabolite and hormone actions that differ by sex. In the present study, males exhibited higher mean DMI, ADG, BW^0.75^, REA, and RF than females. In addition, the growth traits and BF were influenced by the interaction of sex with RFI or RWI class. The difference observed between sexes for carcass traits agrees with the literature, showing a smaller REA and greater BF in females compared to males [10,36,37]. The differences in carcass traits between sexes can be explained by the fact that males have a higher anabolic rate of muscle tissue deposition than females, in addition to late adipose tissue deposition [38,39]. Additionally, heavier animals tend to produce carcasses that are heavier and that provide a higher yield [40], as demonstrated in our study by the correlation of BWi with REA, BF, and RF. 

Similar to our findings, Sakowski et al. [41] reported that bulls had a lower proportion of fat (BF) in their body composition than heifers, showing that the trends of fat deposition in heifers’ is more intense. Lower feed conversion and a higher rate of growth in males have been associated with their better performance over females, as a consequence of the effects of testosterone in males [40]. The higher weight in males can be associated with a higher diameter of the longissimus dorsi muscle, and the fat deposition was associated with leptin activity in the growing phase [42]. 

On the other hand, the FR was lower in males, a finding that might be related to the differences observed in most feeding and drinking behavior traits studied. In the feed conversion tests, the males visited the feed bunk and water troughs less often than females. Additionally, the visits with intake were also lowest compared to the females. The regulation of ingestive behavior and energy expenditure depends on multiple organs and is influenced by sex hormones. Therefore, those findings could be associated with differences in the maintenance of energy balance and ingestive metabolism between females and males [43].

The differences in growth traits between females and males have been widely reported in the literature. The Caracu herd studied herein has participated in a breeding program since 1978, and the selection is based on weight gain, contributing to the differences observed between sexes. The action of androgen hormones promotes muscle hypertrophy in males, which have a larger number of muscle fibers and consequently a higher birth weight compared to females [44,45]. This higher weight of males is observed throughout the animal’s life [46,47]. Similar results were with Nellore, where males consumed 1.08 kg more dry matter than females and consequently had a higher ADG (0.27 kg/d) [48]. Analyzing the weight gain of Pantaneiro cattle, Brito et al. [49] observed a 10% higher asymptotic weight (mature weight) in males compared to females (191.2 kg and 173.5 kg, respectively). Furthermore, Mueller et al. [50] reported differences in performance between sexes in Angus. These differences are intrinsically linked to the WI and feeding behaviors of these animals during growth. It suggests that animals with low RFI, low RWI, and high performance can be selected through the observation of water and feed efficiency associated with ingestive behavior and growth performance. 

Both high RFI and low RFI animals differed in terms of DMI and FD, which is consistent with the results reported for Angus cattle at 17 months of age [51], for Charolais, Hereford, Limousin, and Angus cattle at 10 months of age [52], and for Caracu steers at 13 months of age [13]. The metabolic weight is related to the body weight of the animal and to its allometric growth [53], corroborating the differences in BW^0.75^ between high RWI and low RWI obtained in our study. 

The environmental temperature and nutritional requirements contribute to the differences in DR found between sexes over the four periods of the day (dawn, morning, afternoon, and night) since DR is associated with sex, age, and climate characteristics [30]. In Brahman cattle, a variation of 7% in the temperature–humidity index increased WI by 13% [54]. The present study demonstrated the association between water and feed requirements of the animals based on the positive correlation of WI with both DMI and BW^0.75^. In addition, low RWI and low RFI animals consumed 0.13 L/min more water than high RWI and high RFI animals. The effect of water on animal performance generally receives little attention in livestock farming. However, the need for the sustainable use of water as well as the increase in environmental temperature and its association with herd efficiency have encouraged research on the subject [25,55].

Low RWI animals consume more water than high RWI animals, whereas low RFI animals consume more feed than high RFI animals, i.e., less feed- and water-efficient animals need to consume more feed and water. Since the WI of an adult bovine ranges from 4 to 10 L/kg of ingested DM (25–35 °C) [27], animals that consume larger amounts of feed need to consume more water, corroborating our findings.

Low RWI animals visited fewer troughs and had a lower frequency of visits with intake. Similar results were obtained for Senepol [56], Angus [9], and crossbred cattle [25]. The study of ingestive behavior is an auxiliary tool that permits adapting management practices and, when combined with feed efficiency, identifying more productive and efficient animals [10]. 

Ingestive behavior is divided into three main phases: feeding, ruminating, and idling [57]. However, the frequencies of these behaviors vary according to age, breed, type of diet, management, reactivity, sex, and environment [58]. The higher FVF and NF of females did not result in higher ADG, DMI, or BW^0.75^, which is also associated with the FD at the feed bunk. These findings showed the differences in food metabolism between males and females and agreed with the higher FR observed here in females compared to males. 

In crossbred cattle (Angus, Simmental, Charolais, and Piedmontese) the daily time and number of daily visits to the feed bunk were similar in low and high RFI animals [14]. Similar results were observed in NF and FD, in which RFI classes were no different in Caracu cattle. Some studies suggest that more efficient animals visit feeders less frequently and exhibit a shorter duration of feeding per day [9,59,60,61]. Our study showed that high RFI animals visited feeders around seven times less than low RFI animals. Therefore, the animal that visits the feeder more often and consumes feed and that spends more time per day feeding at the bunk does not always exhibit greater weight gain. This fact demonstrates that feed efficiency is a complex trait influenced by various factors and that knowledge about animal behavior can also contribute to the selection of efficient animals [8,47,62].

The lower FVF observed in males was associated with the smaller number of feed bunkers visited, but the FD was similar between sex. According to Segabinazzi et al. [63], male Holstein calves stayed in feeders for 171 min, a time similar to the observed in our study for Caracu males (162 min). The weight of the animal and its ingestive behavior are associated with the maintenance energy necessary for performing daily activities [64]. A lighter animal may need more energy for maintenance per unit of gain, due to its fast metabolism, and more frequent search for food, which does not necessarily characterize it as an efficient animal in terms of weight gain or good carcass yield.

The ingestive behavior is also directly related to the social hierarchy that is established through dominant relationships, with social position affecting several behaviors such as feed and water intake [65,66,67]. As described by Deniz et al. [68], the social position of a cow influences the time at the feed bunk and is determined by its body mass, body length and age, and social position. Although not evaluated in the present study, displacement at the feeder was frequently observed, especially at the times when the feeders were filled. Haskell et al. [69] found DMI to be affected by dominance in finishing beef steers, which agrees with the high positive correlation between DMI and BWi observed in our study. Thus, it is clear that social dominance also affects feed efficiency through its influence on feeding behavior. Increasing the feeding space can reduce competition, which will improve the feeding behavior of cattle, benefitting animals with small body mass and body length, which are subordinate animals.

The animals spent about 20 min per day drinking water. This duration is longer than the 10 min reported for Holstein animals [63]. This difference might be explained by the fact that the study by Segabinazzi et al. [63] was conducted in southwestern Paraná, Brazil, a region characterized by a temperate climate with milder temperatures than those found in the region of Sertaozinho (SP) where the present study was carried out. Thus, the differences in ingestive behavior might be associated with the climate of the region, animal age, breed, and particularly with the type of diet [70,71].

The behavior traits related to WI (DR, NW, and FVW) observed in both sexes were similar to those obtained for the behavior traits related to feed intake, demonstrating that nutritional, metabolic, and hormonal differences between males and females interfere with these traits [58]. Cattle are gregarious animals that establish dominance in a group; thus, each individual can modify its behavior according to the presence of another animal, which is strongly influenced by the action of the dominant animal [72]. Females tend to form small groups, whereas males tend to be more solitary [72,73]. We, therefore, believe that, following the behavior of the dominant female, the animals visited the trough but did not consume because they were satiated.

Animal feed efficiency is influenced by the consumption rate (DMI), which is highly associated with WI and ingestive behaviors showing an important role in growth performance and development. These factors can be used in animal breeding programs to increase herd efficiency and reduce livestock water use, thus contributing to sustainable beef cattle production.

## 5. Conclusions

The lesser frequency of visits to feeders associated with feeding duration and efficient use of feed were traits important to consider in highly efficient animals. The variability between animals in the herd showed that the animal that visits the feeder more often, consumes feed, and spends more time per day feeding at the bunk does not always exhibit greater weight gain. The RWI trait can be incorporated into selection objectives, showing an important role in the performance traits of the animal. The sexual dimorphism associated with feed efficiency (RFI) and water efficiency (RWI) in Caracu cattle interferes with animal behavior, as well as its growth and development, being important factors to be considered in livestock production.

## Figures and Tables

**Table 1 animals-12-03196-t001:** Pearson correlation among feed efficiency, ingestive behavior, and carcass traits in males (above the diagonal) and in females (below the diagonal).

	BWi	ADG	RFI	DMI	BW^0.75^	REA	BF	RF	WI	RWI	NW	FVW	DD	NF	FVF	FD
**BWi**	−	0.19 ^NS^	0.03 ^NS^	0.68 ***	0.98 ***	0.61 ***	0.15 ^NS^	0.33 **	0.47 ***	−0.42 **	0.01 ^NS^	−0.09 ^NS^	0.26 *	−0.26 *	−0.44 ***	0.22 ^NS^
**ADG**	0.07 ^NS^	1.00	0.12 ^NS^	0.63 ***	0.27 *	0.28 *	0.01 ^NS^	−0.09 ^NS^	0.49 ***	0.25 *	0.23 ^NS^	0.19 ^NS^	0.08 ^NS^	0.06 ^NS^	0.07 ^NS^	−0.07 ^NS^
**RFI**	−0.07 ^NS^	−0.25 ^NS^	1.00	0.58 ***	0,06 ^NS^	0.13 ^NS^	0.28 *	0.03 ^NS^	0.11 ^NS^	0.01 ^NS^	0.18 ^NS^	0.26 *	0.01 ^NS^	0.25 ^NS^	0.42 **	0.06 ^NS^
**DMI**	0.69 ***	0.31 *	0.52 **	1.00	0.74 ***	0.55 ***	0.27 *	0.19 ^NS^	0.60 ***	−0.08 ^NS^	0.20 ^NS^	0.19 ^NS^	0.24 ^NS^	−0.01 ^NS^	−0.03 ^NS^	0.13 ^NS^
**BW^0.75^**	0.98 ***	0.17 ^NS^	0.10 ^NS^	0.73 ***	1.00	0.62 ***	0.20 ^NS^	0.35 **	0.56 ***	−0.34 **	0.03 ^NS^	−0.05 ^NS^	0.31 **	−0.26 *	−0.46 ***	0.23 ^NS^
**REA**	0.75 ***	0.04 ^NS^	−0.05 ^NS^	0.53 **	0.75 ***	1.00	0.21 ^NS^	0.13 ^NS^	0.23 ^NS^	−0.35 **	−0.01 ^NS^	−0.08 ^NS^	0.19 ^NS^	−0.11 ^NS^	−0.22 ^NS^	0.28 *
**BF**	0.39 **	−0.06 ^NS^	−0.01 ^NS^	0.26 ^NS^	0.39 **	0.26 ^NS^	1.00	0.43 ***	0.15 ^NS^	−0.02 ^NS^	0.16 ^NS^	0.26 *	0.35 **	0.02 ^NS^	−0.02 ^NS^	0.22 ^NS^
**RF**	0.33 *	−0.01 ^NS^	−0.13 ^NS^	0.17 ^NS^	0.36 **	0.37 **	0.54 ***	1.00	0.11 ^NS^	−0.21 ^NS^	−0.06 ^NS^	0.04 ^NS^	0.35 **	−0.13 ^NS^	−0.15 ^NS^	0.08 ^NS^
**WI**	0.55 ***	0.17 ^NS^	0.06 ^NS^	0.54 ***	0.60 ***	0.36 **	0.08 ^NS^	0.12 ^NS^	1.00	0.59 ***	0.28 **	0.23 ^NS^	0.27 *	−0.04 ^NS^	−0.23 ^NS^	−0.03 ^NS^
**RWI**	−0.18 ^NS^	0.05 ^NS^	0.11 ^NS^	−0.01 ^NS^	−0.14 ^NS^	−0.21 ^NS^	−0.24 ^NS^	−0.16 ^NS^	0.71 ***	1.00	0.28 *	0.30 *	0.01 *	0.20 ^NS^	0.16 ^NS^	−0.25 *
**NW**	−0.39 **	−0.13 ^NS^	0.11 ^NS^	0.29 ^NS^	−0.44 **	−0.36 **	−0.45 **	−0.26 ^NS^	−0.12 ^NS^	0.23 ^NS^	1.00	0.76 ***	0.27 **	0.47 **	0.17 ^NS^	−0.09 ^NS^
**FVW**	−0.39 **	−0.05 ^NS^	0.04 ^NS^	0.30 ^NS^	−0.41 **	−0.19 ^NS^	−0.48 **	−0.30 *	−0.10 ^NS^	0.24 ^NS^	0.76 ***	1.00	0.51 ***	0.39 **	0.22 ^NS^	−0.09 ^NS^
**DD**	0.11 ^NS^	−0.34 *	−0.06 ^NS^	−0.12 ^NS^	−0.07 ^NS^	0.25 ^NS^	−0.07 ^NS^	−0.06 ^NS^	0.13 ^NS^	0.12 ^NS^	0.33 *	0.33 *	1.00	0.08 ^NS^	−0.14 ^NS^	0.17 ^NS^
**NF**	−0.43 **	−0.03 ^NS^	0.15 ^NS^	−0.21 ^NS^	−0.42 **	−0.50 **	−0.02^NS^	−0.24 ^NS^	−0.18 ^NS^	0.13 ^NS^	0.24 ^NS^	0.19 ^NS^	−0.17 ^NS^	1.00	0.48 ***	0.01 ^NS^
**FVF**	−0.37 **	0.19 ^NS^	0.30 ^NS^	−0.01 ^NS^	−0.37 *	−0.22 ^NS^	0.02^NS^	−0.13 ^NS^	−0.28 ^NS^	−0.05 ^NS^	0.21 ^NS^	0.19 ^NS^	−0.08 ^NS^	0.54 ***	1.00	−0.24 ^NS^
**FD**	0.12 ^NS^	−0.05 ^NS^	−0.23 ^NS^	−0.05 ^NS^	0.14 ^NS^	0.32 *	0.06^NS^	0.15 ^NS^	0.04 ^NS^	−0.06 ^NS^	−0.06 ^NS^	0.13 ^NS^	0.24 ^NS^	−0.23 ^NS^	−0.24 ^NS^	1.00

*: *p* < 0.05, **: *p* < 0.01, ***: *p* < 0.0001, ^NS^: non-significant; BWi: initial body weight; ADG: average daily gain (kg/d); RFI: residual feed intake; DMI: dry matter intake (kg/d); BW^0.75^: mid-test metabolic weight (kg); REA: rib eye area (cm^2^); BF: backfat thickness (mm); RF: rump fat thickness (mm); WI: water intake (L/d); RWI: residual water intake; NW: number of water troughs visited per day (number); FVW: frequency of visits to the water through with intake (visits/d); DD: drinking duration (min/d); NF: number of feed bunks visited per day (number); FVF: frequency of visits to the feed bunk with intake (visits/d); FD: feeding duration (min/d).

**Table 2 animals-12-03196-t002:** Mean and standard error of dry matter intake (DMI), water intake (WI), average daily gain (ADG), mid-test metabolic body weight (BW^0.75^), rib eye area (REA), backfat thickness (BF), and rump fat thickness (RF) according to the effects of sex, residual feed intake (RFI), and residual water intake (RWI) class.

Trait	Sex			RFI			RWI		
	Male (*n* = 61)	Female (*n* = 43)	*p*	Low (*n* = 47)	High (*n* = 57)	*p*	Low (*n* = 47)	High (*n* = 57)	*p*
DMI (kg/d)	9.14 ± 0.12	7.33 ± 0.18	0.0001	8.59 ± 0.09	7.87 ± 0.09	0.0001	8.27 ± 0.08	8.19 ± 0.11	0.6032
WI (L/d)	21.16 ± 0.44	19.36 ± 0.67	0.0744	20.62 ± 0.34	19.91 ± 0.37	0.1292	22.21 ± 0.31	18.32 ± 0.41	0.0001
ADG (kg/d)	1.21 ± 0.03	0.72 ± 0.04	0.0001	0.98 ± 0.03	0.96 ± 0.02	0.6538	1.01 ± 0.02	0.93 ± 0.03	0.0650
BW^0.75^ (kg)	70.37 ± 0.99	59.47 ± 1.46	0.0001	64.76 ± 0.74	65.08 ± 0.81	0.7540	64.52 ± 0.69	65.32 ± 0.90	0.4663
REA (cm^2^)	68.84 ± 1.23	39.38 ± 1.82	0.0001	54.27 ± 0.93	53.98 ± 1.01	0.7999	53.57 ± 0.86	54.65 ± 1.13	0.4349
BF (mm)	2.34 ± 0.24	2.85 ± 0.36	0.3531	2.78 ± 0.18	2.40 ± 0.20	0.1287	2.40 ± 0.17	2.79 ± 0.22	0.1597
RF (mm)	5.21 ± 0.21	3.66 ± 0.31	0.0016	4.41 ± 0.16	4.47 ± 0.17	0.7787	4.41 ± 0.15	4.46 ± 0.19	0.8639

**Table 3 animals-12-03196-t003:** Least square means of the feeding rate (FR) and drinking rate (DR) of males and females according to period (dawn, morning, afternoon, and night), residual feed intake (RFI), residual water intake (RWI), interaction of sex and period, RFI, and RWI class. Different superscript letters denote significant differences (*p* < 0.05).

Trait	FR (kg/min)	*p*-Value	DR (L/min)	*p*-Value
**Sex**		0.0001		0.0005
Male	0.08 ± 0.001 ^b^		1.05 ± 0.03 ^b^	
Female	0.09 ± 0.001 ^a^		0.90± 0.03 ^a^	
**Period**		0.0001		0.0502
Dawn	0.06 ± 0.002 ^d^		0.89 ± 0.04 ^b^	
Morning	0.10 ± 0.002 ^a^		1.03 ± 0.04 ^a^	
Afternoon	0.09 ± 0.002 ^b^		0.97 ± 0.04 ^a,b^	
Night	0.08 ± 0.002 ^c^		1.01 ± 0.04 ^a,b^	
**RFI**		0.0152		0.0163
High	0.089 ± 0.001 ^a^		0.92 ± 0.03 ^a^	
Low	0.085 ± 0.001 ^b^		1.05 ± 0.03 ^b^	
**RWI**		0.2121		0.0033
High	0.086 ± 0.001		0.91 ± 0.03 ^b^	
Low	0.089 ± 0.001		1.04 ± 0.03 ^a^	
**Sex * Period**		0.0919		0.0001
Male * Dawn	0.06 ± 0.002		1.05 ± 0.05 ^a,b^	
Male * Morning	0.10 ± 0.002		1.01 ± 0.05 ^a,b^	
Male * Afternoon	0.09 ± 0.002		0.97 ± 0.05 ^a,b,c^	
Male * Night	0.08 ± 0.002		1.19 ± 0.05 ^a^	
Female * Dawn	0.06 ± 0.003		0.73 ± 0.06 ^c^	
Female * Morning	0.11 ± 0.003		1.06 ± 0.06 ^a,b^	
Female * Afternoon	0.10 ± 0.003		0.98 ± 0.06 ^a,b,c^	
Female * Night	0.09 ± 0.003		0.83 ± 0.06 ^b,c^	
**Sex * RFI**		0.0197		0.3030
Male * High	0.084 ± 0.002 ^b^		1.03 ± 0.04	
Male * Low	0.083 ± 0.001 ^b^		1.08 ± 0.03	
Female * High	0.095 ± 0.002 ^a^		0.83 ± 0.05	
Female * Low	0.087 ± 0.002 ^b^		0.97 ± 0.04	
**Sex * RWI**		0.8029		0.8337
Male * High	0.083 ± 0.002		0.98 ± 0.03	
Male * Low	0.085 ± 0.001		1.13 ± 0.04	
Female * High	0.089 ± 0.002		0.84 ± 0.06	
Female * Low	0.092 ± 0.002		0.96 ± 0.03	

* Interaction between traits.

## Data Availability

The data presented in this study are available on request from the corresponding author. The data are not publicly available due to privacy or ethical restrictions, and the data that support the findings of this study are available in the Appendix A of this article.

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
