# Peer review of "Effects of Feeding and Drinking Behavior on Performance and Carcass Traits in Beef Cattle"

_animals, 2022, doi:10.3390/ani12223196_

Round 1

Reviewer 1 Report

Dear Editor,

Despite the fact that the study was well planned, designed, and implemented, the writing process showed a number of serious deficiencies. In particular, the section in which the results are presented is in need of serious revision. It should be noted that the results are not arranged in a specific order, nor are they dependent on some tables displaying the results stated in the text. The tables do not contain most of the values referenced in the text. Therefore, it was not possible to conduct a healthy evaluation in the “discussion” and “conclusion” sections since there are serious problems with the results section of the text.

Major comments: 

A certain order must be maintained in the results section, but the authors presented the data in a disorganized manner. As an example, Table 2 is presented prior to Table 1.

L117-129: It is recommended that the equation part of the word be used for formula.

L131, L136: It is recommended to rewrite the descriptions in L178-183 in order to add an appropriate abbreviation for L131 and L136. As an example, authors may define "low RFI" and "high RFI" (which were given within parentheses in L178-179) in L131. The authors use the terms “most efficient” and “less efficient” in conjunction with “high-efficiency” and “low-efficiency” in the same sentence, which is confusing. Therefore, it will be more understandable if this part is written using an abbreviation for “high efficiency RFI” and “low efficiency RFI”.

L178-179: The values of 8.67 ± 0.08kg/d and 7.94 ± 0.09kg/d given in L178-179 need to be checked.

L180-183: What are the "most efficient animals"? (according to what, feed consumption, or water consumption).

L183-184: The data in Figure 1 and Table 2 should be presented in separate paragraphs, and the prominent data in Figure 1 should be highlighted.

L185: In Figure 1, the thicknesses of the lines selected for WI, DMI, and FVM are greater than the thicknesses of the columns of REA and RVF. A more understandable shape may emerge if the shape is colored and especially if the line thickness of the FVM is reduced. However, some distributions cannot be observed precisely due to the thickness of the FVM. It would be better if the y-axis numbers did not require a decimal point.

L190-211: It would be better to insert the correlation coefficient in parentheses in L190-211, where the results of Table 1 are provided, as in "RFI and BF (0.28)".

L221-222:  FD and DR are not mentioned in Table 2. How did this conclusion come to be reached?

L224-225:  The importance of this difference should be emphasized by giving p value.

L225-229: In the text and supplementary file, please indicate from which table the ones between L225-229 were accessed.

L230-239: The values and significance levels described in these lines are not found anywhere else in the text. The interaction between sex and RWI cannot be controlled in a table.

L245-249: It is recommended that the values in parentheses be checked.

L254-257: There is a need to verify the significance levels in parentheses. The p value presented in Table 3 is contradictory.

L258-264: There is a need to rewrite this paragraph so as not to contradict the values given in Table 3.

L265-271: The information provided in this paragraph could not be controlled since descriptive statistics by sex are not included in Supplementary Table S2. Therefore, there is a need to revise this paragraph and Supplementary Table S2.

L272-274: There is no table that contains the values in parentheses.

L275-284: There is no table that contains the values and significance levels in parentheses.

L287: In table 3, it is recommended that the lettering of RFI be checked. The differences were not significant.

Minor comments:

L33: Since the abbreviation "BF" is no longer used in the Abstract section, it should be removed.

L46, L181, L370, L380, L434 : “WI” instead of “water intake”.

L60-61: Citations should be provided for several studies involving this variation.

L77: “backfat thickness (BF)” instead of “BF”.

L84: “Creole” instead of “creole”.

L90: “cattle “ instead of “animals”.

L108: “dry matter (DM)”  instead of “dry matter”.

L110, L111, L114, L353, L381: “DM” instead of “dry matter”.

L132: “Water intake (L/d)” instead of “Water intake (WI; L/d)”. There was an earlier abbreviation of water intake in L43

L148: “rib eye area (REA)” instead of “REA”. It was previously used the abbreviations BF and RF; however, the abbreviation REA was first used in the text.

L152-L153: “12th and 13th” instead of “12th and 13th”. Using a superscript is recommended.

L176, L216: “cm2” instead of “cm2”.

L198, L199, L201, L202, L205: “BW0.75” instead of “BW0.75”.

L199, L200: “ ~ ” should be removed.

L213: It is recommended to take into account the column widths of Table 1

L214: “*: P < 0.05, **: P < 0.01, ***: P < 0.001” instead of “*P = 0.05, **P = 0.01, ***P = 0.0001”.

L217-219: Units of RWI, NW, and NF, as well as others, should be specified in parentheses.

L221: "(P > 0.05)" should be placed at the end of the sentence as “(P < 0.001)”.

L224: “Females and males showed similar WI (P = 0.0744) and BF (0.3531)” instead of “Females and males showed similar WI (P = 0.0744)”.

L243: The “n” numbers should be given in parentheses in Table 2. Further, there should be a slight increase in the width of the columns with the P value.

L250-251: An indication of the level of significance should be included at the end of the sentence.

L251-253: It is important to note, however, that the difference is not significant.

Author Response

Thank you for the opportunity to revise our manuscript. We appreciate your precious time in reviewing our paper and providing valuable comments that led to possible improvements in the current version. The authors have carefully considered the comments and tried our best to address every one of them.

Major comments:

A certain order must be maintained in the results section, but the authors presented the data in a disorganized manner. As an example, Table 2 is presented prior to Table 1.

Authors: Thanks for pointing that out. The results section has been organized as requested.

L117-129: It is recommended that the equation part of the word be used for formula.

Authors: Thank you for your suggestion. Actually, the equation part of the word has been used for formula in this manuscript. I think that occurred some problems with the MS Word version used (L120, L126, L129).

L131, L136: It is recommended to rewrite the descriptions in L178-183 in order to add an appropriate abbreviation for L131 and L136. As an example, authors may define "low RFI" and "high RFI" (which were given within parentheses in L178-179) in L131. The authors use the terms “most efficient” and “less efficient” in conjunction with “high-efficiency” and “low-efficiency” in the same sentence, which is confusing. Therefore, it will be more understandable if this part is written using an abbreviation for “high efficiency RFI” and “low efficiency RFI”.

Authors: Thank you for your suggestion. The sentence has been rewritten (L225- L230).

L178-179: The values of 8.67 ± 0.08kg/d and 7.94 ± 0.09kg/d given in L178-179 need to be checked.

Authors: Thanks for pointing that out. The values were checked (L225-L226).

L180-183: What are the "most efficient animals"? (according to what, feed consumption, or water consumption).

Authors: The sentence has been rewritten (L225- L230).

L183-184: The data in Figure 1 and Table 2 should be presented in separate paragraphs, and the prominent data in Figure 1 should be highlighted.

Authors: Done.

L185: In Figure 1, the thicknesses of the lines selected for WI, DMI, and FVM are greater than the thicknesses of the columns of REA and RVF. A more understandable shape may emerge if the shape is colored and especially if the line thickness of the FVM is reduced. However, some distributions cannot be observed precisely due to the thickness of the FVM. It would be better if the y-axis numbers did not require a decimal point.

Authors: Figure 1 was removed.

L190-211: It would be better to insert the correlation coefficient in parentheses in L190-211, where the results of Table 1 are provided, as in "RFI and BF (0.28)".

Authors: Done (L189-215).

L221-222:  FD and DR are not mentioned in Table 2. How did this conclusion come to be reached?

Authors: The results of DR were shown in Table 3. The FD is an ingestive behavior trait and its results are shown in the main text and now these results can be also found in Supplementary Figure 4.

L224-225:  The importance of this difference should be emphasized by giving p value.

Authors: The P value (P = 0.0001) was included in the main text (L235).

L225-229: In the text and supplementary file, please indicate from which table the ones between L225-229 were accessed.

Authors: Thank you for your suggestion. The age of the animals was inserted in the model with a covariate and it is a numerical variable (L243). These results are presented in Supplementary Figure 1 (FS1).

L230-239: The values and significance levels described in these lines are not found anywhere else in the text. The interaction between sex and RWI cannot be controlled in a table.

Authors: We presented the results of the interaction between sex and RWI for the DMI, BW0.75, and BF traits only in the main text. Now, these results could be verified in Supplementary Figure 2 (FS2).

L245-249: It is recommended that the values in parentheses be checked.

Authors: These values were obtained by the interaction between sex and RWI. The results of the interaction between sex and RWI were described only in the main text (L301-L305) and now these results are presented in Supplementary Figure 2 too (FS2). The values obtained by the sex and RFI interaction were shown in Supplementary Figure 3. These results are different from those exhibited in Table 2 because Table 2 has only the descriptive statistics (L257-L261).

L254-257: There is a need to verify the significance levels in parentheses. The p value presented in Table 3 is contradictory.

Authors: Thanks for pointing that out. The values were checked (L267).

L258-264: There is a need to rewrite this paragraph so as not to contradict the values given in Table 3.

Authors: In this paragraph, we presented the difference in DR between males and females (1.05 and 0.90 L/min, respectively). After, we presented the results of the interaction between sex and period of the day, whereby males exhibited higher DR at night and females exhibited lower DR at dawn. No difference was observed between sex in the morning and in afternoon periods. Finally, we presented the results of DR according to RFI and RWI classes. The values and P-values presented in the text are in accordance with Table 3 (L271-L277).

L265-271: The information provided in this paragraph could not be controlled since descriptive statistics by sex are not included in Supplementary Table S2. Therefore, there is a need to revise this paragraph and Supplementary Table S2.

Authors: The results of behavior traits at the feed bunks (NF, FVF, and FD) and at the water troughs (NW, FVW, and DD) according to sex, RFI class, or RWI class and their interaction have been described only in the main text. Now, we included Supplementary Figure 4 (FS4), where these results could be checked.

L272-274: There is no table that contains the values in parentheses.

Authors: These values could be verified in Supplementary Figure 4 (FS4) and Supplementary Figure 5 (FS5).

L275-284: There is no table that contains the values and significance levels in parentheses.

Authors: Now, we included these results in Supplementary Figure 4 (FS4) and Supplementary Figure 5 (FS5). 

L287: In table 3, it is recommended that the lettering of RFI be checked. The differences were not significant.

Authors: The results showed that the RFI effect influences the FR and DR, with P-value equal to 0.0152 and 0.0163 respectively (without interaction). The differences are significant because we considered P < 0.05. The interaction between sex and RFI influenced the FR (P=0.0197), however, this interaction did not influence the DR (P=0.3030).

Minor comments:

L33: Since the abbreviation "BF" is no longer used in the Abstract section, it should be removed.

Authors: Done (L33).

L46, L181, L370, L380, L434 : “WI” instead of “water intake”.

Authors: Done.

L60-61: Citations should be provided for several studies involving this variation.

Authors: We included four citations (L61).

L77: “backfat thickness (BF)” instead of “BF”.

Authors: Done.

L84: “Creole” instead of “creole”.

Authors: Done.

L90: “cattle “ instead of “animals”.

Authors: Done.

L108: “dry matter (DM)”  instead of “dry matter”.

Authors: Done.

L110, L111, L114, L353, L381: “DM” instead of “dry matter”.

Authors: Done.

L132: “Water intake (L/d)” instead of “Water intake (WI; L/d)”. There was an earlier abbreviation of water intake in L43

Authors: Done.

L148: “rib eye area (REA)” instead of “REA”. It was previously used the abbreviations BF and RF; however, the abbreviation REA was first used in the text.

Authors: Done (L159).

L152-L153: “12th and 13th” instead of “12th and 13th”. Using a superscript is recommended.

Authors: Done (L163).

L176, L216: “cm2” instead of “cm2”.

Authors: Done (L187, L220).

L198, L199, L201, L202, L205: “BW0.75” instead of “BW0.75”.

Authors: Done.

L199, L200: “ ~ ” should be removed.

Authors: Done (L201).

L213: It is recommended to take into account the column widths of Table 1

Authors: The table is too large and the minimum font size was applied (7 pt).

L214: “*: P < 0.05, **: P < 0.01, ***: P < 0.001” instead of “*P = 0.05, **P = 0.01, ***P = 0.0001”.

Authors: Thanks for pointing that out. Done (L218).

L217-219: Units of RWI, NW, and NF, as well as others, should be specified in parentheses.

Authors: Thanks for pointing that out. RWI is a residual of the regression equation (without unit) (L221-L224).  

L221: "(P > 0.05)" should be placed at the end of the sentence as “(P < 0.001)”.

Authors: Thanks for pointing that out. (L232).

L224: “Females and males showed similar WI (P = 0.0744) and BF (0.3531)” instead of “Females and males showed similar WI (P = 0.0744)”.

Authors: Done (L234).

L243: The “n” numbers should be given in parentheses in Table 2. Further, there should be a slight increase in the width of the columns with the P value.

Authors: The n was included as a footnote (L244).

L250-251: An indication of the level of significance should be included at the end of the sentence.

Authors: Done (L264).

L251-253: It is important to note, however, that the difference is not significant.

Authors: Thanks for pointing that out. The sentence has been modified (L263-L264).

Reviewer 2 Report

I have gone through the manuscript. The results of this study will contribute to a better understanding of the effects of ingestive behavior and growth traits. The manuscript is well written, and I would like to suggest minor revision.

Line 89-147: In the “Materials and Methods” section the feed efficiency test should be described in more detail. The data were collected by the electric devices or monitored through direct observation? Please add any references. The following documents or others may be helpful.

J. Dairy Sci. 90:5732–5736   doi:10.3168/jds.2007-0331

Line 90: The males cattle evaluated had been already castrated?

Line 131, 135: Are these classification into two categories common? The descriptive statistics of all data and each class, especially RFI and RWI, should be shown in the Table. 

Line 183: This sentence is slightly confusing, so please describe it in more detail. That means “All the animals with low RWI exhibited low RFI, and all the animals with high RWI exhibited high RWI”? If so, why was the Pearson correlation between RFI and RWI not significant (Table 1)? Please describe in the “Discussion” section.

Line 185 (Figure 1): Is the horizontal axis the ID of the animal? There are no numbers on the horizontal axis in the figure above. Lines showing DMI and FVW are broken and hard to see, and the dotted line seems to be the data for each individual. 

Line 415-426: Generally, beef cattle seem to be raised in herds such as feedlots. Since this study is a single herd, there was no need to consider the social hierarchy; hence further study is needed to improve the environmental sustainability of beef cattle production. Please add these sentences.

Author Response

Thank you for the opportunity to revise our manuscript. We appreciate your precious time in reviewing our paper and providing valuable comments that led to possible improvements in the current version. The authors have carefully considered the comments and tried our best to address every one of them.

Line 89-147: In the “Materials and Methods” section the feed efficiency test should be described in more detail. The data were collected by the electric devices or monitored through direct observation? Please add any references. The following documents or others may be helpful. J. Dairy Sci. 90:5732–5736   doi:10.3168/jds.2007-0331

Authors: Thank you for your suggestion. More details of the feed efficiency evaluation were included (L94-97) and some references were cited (L96, L152). The data of feeding and drinking behavior were also obtained by the Intergado® electronic feed bunks and drinkers (L151-L158).

Line 90: The males cattle evaluated had been already castrated?

Authors: The males used in this study were not castrated (intact males). This information is included in the manuscript (L90).

Line 131, 135: Are these classifications into two categories common? The descriptive statistics of all data and each class, especially RFI and RWI, should be shown in the Table. 

Authors: The classification of RWI and RFI into two classes is very useful in the literature (DOI: 10.1093/jas/skz354, 10.1016/j.meatsci.2017.02.004, 10.1371/journal.pone.0272236). The descriptive statistics of RFI and RWI by classes were included in Supplementary Table 2, as requested.

Line 183: This sentence is slightly confusing, so please describe it in more detail. That means “All the animals with low RWI exhibited low RFI, and all the animals with high RWI exhibited high RWI”? If so, why was the Pearson correlation between RFI and RWI not significant (Table 1)? Please describe in the “Discussion” section.

Authors: Thanks for pointing that out. I am sorry for this mistake. The sentence was deleted.

Line 185 (Figure 1): Is the horizontal axis the ID of the animal? There are no numbers on the horizontal axis in the figure above. Lines showing DMI and FVW are broken and hard to see, and the dotted line seems to be the data for each individual. 

Authors: Figure 1 was removed.

Line 415-426: Generally, beef cattle seem to be raised in herds such as feedlots. Since this study is a single herd, there was no need to consider the social hierarchy; hence further study is needed to improve the environmental sustainability of beef cattle production. Please add these sentences.

Authors: Thank you for your suggestion. Actually, beef cattle production in Brazil is predominantly an extensive grazing system and several studies have described social hierarchy in a single herd (DOI: 10.1016/j.applanim.2021.105390, 10.3389/fvets.2020.00061).

Round 2

Reviewer 1 Report

Dear authors,

I would like to carefully review your revised article with the id number "animals-2024325". In my opinion, this revised article incorporates all of the points raised in the original draft to the fullest extent possible. However, it would be helpful if you could make one final minor correction to the text according to the suggestions given below. In addition to this small point, I believe you have filled a significant gap in the literature regarding the efficiency of feed and water in beef cattle production, which is understudied both economically and environmentally. Congratulations to each of the authors who contributed to this beautiful work and best wishes for their future endeavors.

L229: Please use “P > 0.05” instead of “P = 0.3531”.

L239: The n numbers in Table 2 would be better stated as male, female, low-RFI, high-RFI, low RWI, high RWI in parentheses rather than “n=104”.

L281: Please use “Figure S4” instead of “Supplementary Figure 4”.

Author Response

Dear reviewer,

We appreciate your precious time in reviewing our manuscript and providing valuable comments that led to possible improvements in the R2 version. The authors have carefully considered the comments.

L229: Please use “P > 0.05” instead of “P = 0.3531”.

Authors: Done (L229).

L239: The n numbers in Table 2 would be better stated as male, female, low-RFI, high-RFI, low RWI, high RWI in parentheses rather than “n=104”.

Authors: Done (L238).

L281: Please use “Figure S4” instead of “Supplementary Figure 4”. 

Authors: Done (L281).